# OsMGT1 Confers Resistance to Magnesium Deficiency By Enhancing the Import of Mg in Rice

**DOI:** 10.3390/ijms20010207

**Published:** 2019-01-08

**Authors:** Ludan Zhang, Yuyang Peng, Jian Li, Xinyue Tian, Zhichang Chen

**Affiliations:** 1Root Biology Center, College of Resources and Environment, Fujian Agriculture and Forestry University, Fuzhou 350002, China; zld_93@163.com (L.Z.); pengyuyang2019@163.com (Y.P.); li123456jian@126.com (J.L.); 2College of Crop Science, Fujian Agriculture and Forestry University, Fuzhou 350002, China; txyacbb@163.com (X.T.)

**Keywords:** OsMGT1, transporter, rice, Mg deficiency

## Abstract

Magnesium (Mg) is an essential nutrient element for plant growth and plays an important role in numerous physiological and biochemical processes. Mg deficiency inhibits plant growth and has become a growing problem for crop productions in agriculture. However, the molecular mechanisms for the resistance to Mg deficiency in plants were not well understood. In this study, we identified a Mg transporter gene *OsMGT1* that confers resistance to Mg deficiency in rice (*Oryza sativa*). The expression of *OsMGT1* was highly induced by Mg deficiency in shoots. Investigation of tissue expression patterns revealed that *OsMGT1* was mainly expressed in the phloem region; however, Mg deficiency remarkably enhanced its expression in xylem parenchyma and mesophyll cells in shoots. Knockout of *OsMGT1* resulted in a significant reduction in Mg content and biomass when grown at Mg-limited conditions. Furthermore, the sensitivity to low-Mg in mutants was intensified by excessive calcium supply. In addition, overexpression of *OsMGT1* increased Mg content and biomass under low-Mg supply. In conclusion, our results indicate that OsMGT1 plays an important role in rice Mg import and is required for the resistance to Mg deficiency, which can be utilized for molecular breeding of low-Mg tolerant plants.

## 1. Introduction

Magnesium (Mg) is an essential element for plant growth, development and reproductive success [1,2,3], which plays an important role in numerous physiological and biochemical processes, such as chlorophyll biosynthesis and degradation, photosynthetic CO_2_ assimilation, carbohydrate allocation, energy metabolism and ribosome aggregation [4,5,6,7,8]. Therefore, lack of Mg in plants reduces the photosynthetic rate, disrupts the distribution of carbohydrates from source to sink, inhibits the growth of plant organs and ultimately leads to a significant decline in crop productivity and quality [9,10]. Mg deficiency in plants may result from three following factors: First, Mg has a relatively larger hydrated radius in contrast to other cations, which makes it easier to be leached, particularly in acidic soils and sandy soils with low cation exchange capacity [11,12,13]. Second, with the increasing crop yield and multi-cropping, soil Mg supply cannot meet crop requirements, resulting in soil Mg depletion [14,15]. Third, the tremendous input of inorganic fertilizers and soil acidification lead to the antagonistic effect of other cations (H^+^, NH_4_^+^, Al^3+^, Mn^2+^) on plant Mg uptake [16]. Therefore, Mg deficiency has become a growing problem for many crop productions in agriculture.

In view of the biological significance and unique chemical property of Mg^2+^, the studies on Mg transporters, which mediate Mg^2+^ uptake, translocation and distribution are increasingly important [17,18]. Cation transporter gene families, such as *MHX* (Mg^2+^/H^+^ Exchanger), *CNGC* (Cyclic Nucleotide-Gated Channel), *HKT* (High-Affinity K^+^ Transport) and *MRS2/MGT* (Mitochondrial RNA Splicing 2/Magnesium Transporter) have been identified as Mg transporters in plants [19,20,21,22,23]. MHX is a unique vacuolar Mg transporter in Arabidopsis. The high expression of *MHX* in vascular tissues suggests its role in xylem loading or retrieval of Mg [19]. CNGCs are commonly known as Ca^2+^-permeable cation transport channels [24]; however, their properties of low cation selectivity suggest that they are also permeable to other cations, including K^+^, H^+^ and Mg^2+^ [25]. OsHKT2;4, a member of the OsHKT2 subfamily with Na^+^-K^+^ symport activity, functions as a low-affinity Mg^2+^ transporter in rice [23]. To date, the *MRS2/MGT* are the best-studied Mg^2+^ transporter gene family in plants [21,26], which are homologs of *CorA* in bacteria and *Alr1* in yeast [27,28,29]. MRS2/MGT proteins form a funnel-shaped homopentamer and individually own two conserved transmembrane domains near their C-terminals [30,31,32]. Cytoplasmic Mg^2+^ is bound between monomers in the cytoplasmic domain for channel gating, while a conserved CorA motif of tripeptide (GMN) which appears at the end of first transmembrane helices, controls ion selectivity [31,32].

So far, the *MRS2/MGT* family has been revealed in several plant species, such as Arabidopsis, rice, soybean and maize [3,20,26,33]. Although most of them have Mg transport activity by functional complementation with yeast and bacteria mutants, the physiological roles in plants are largely different [21,26,34]. AtMGT6 and OsMGT1 are able to mediate Mg uptake in the roots of Arabidopsis and rice, respectively [35,36]. AtMGT6 confers both low- and high-Mg tolerance [36,37], whereas OsMGT1 mediates both Al and salt tolerance [35,38]. *OsMGT2* and *OsMGT6* in rice and *AtMGT9* in Arabidopsis are mainly expressed in root vascular tissues, which are likely to be involved in the xylem loading during Mg translocation from roots to shoots [3,21]. AtMGT10 locates at the chloroplast envelope membrane for regulating Mg homeostasis in chloroplasts, which is crucial for chloroplast development, particularly under high light conditions [39,40,41]. Two mesophyll-abundant and tonoplast-localized transporters AtMGT2 and AtMGT3 are required for high vacuolar Mg storage through transport of Mg into vacuoles [42]. In addition, pollen development and male fertility require plenty of Mg influx, which are facilitated by *AtMGT4*, *AtMGT5* and *AtMGT9* in Arabidopsis [17,43,44,45].

There are 9 *MGT* homologs in the rice genome, but only one of them (*OsMGT1*) has been functionally studied [35,38]. Our previous studies have revealed that OsMGT1 is a plasma membrane-localized transporter, which is highly expressed in root tips and vascular tissues. Knockout of *OsMGT1* results in decreased Mg uptake in the roots by a stable isotope ^25^Mg uptake experiment [35]. This evidence indicates that OsMGT1 is a transporter for root Mg uptake in rice. Furthermore, increasing Mg concentrations in the cytosol by *OsMGT1* contributes to both higher Al and salt tolerance in rice [35,38], indicating diverse roles of *OsMGT1* in response to abiotic stresses. However, whether *OsMGTs* in rice is involved in low-Mg tolerance is unknown. In this study, we firstly investigated the gene expression of all *OsMGTs* in both shoots and roots of rice, and observed that only the expression of *OsMGT1* in the shoots was remarkably induced by Mg deficiency. Knockout of *OsMGT1* resulted in much lower Mg accumulation and higher sensitivity to Mg deficiency, while overexpression of *OsMGT1* enhanced the tolerance to Mg deficiency. Taken together, our results suggest that *OsMGT1* plays an important role in rice growth under low-Mg stress.

## 2. Results

### 2.1. OsMGT1 Was Up-Regulated by Mg Deficiency

We removed Mg from nutrient solution in order to examine the response of *OsMGTs* to Mg deficiency in rice. Real-time RT-PCR results revealed that most of *OsMGTs* in rice have little response to Mg deficiency (Figure 1a). Among nine members, only *OsMGT1* was significantly induced by Mg deficiency (Figure 1a). We found that its expression in the shoots was up-regulated by about 4 times after exposure to Mg deficiency for 7 days, whereas that in the roots was unaffected (Figure 1a). Analysis of shoot spatial expression showed that *OsMGT1* in both leaf blade and leaf sheath was up-regulated in the absence of Mg, but recovered rapidly after the addition of Mg for 24 h (Figure 1b). A time-course experiment showed that the induction of *OsMGT1* occurred at the fifth day after exposure to Mg deficiency and the expression kept a relatively high level after Mg induction (Figure 1c). Furthermore, the expression of *OsMGT1* also can be enhanced by excessive calcium (Ca) supply under −Mg condition (Figure 1d).

### 2.2. Mg Deficiency Altered the Tissue Expression Pattern of OsMGT1

To examine the tissue and cell specificity of *OsMGT1* expression in response to Mg deficiency, we performed immunostaining of the transgenic rice carrying the 2.5 kb promoter sequence of *OsMGT1* fused with green fluorescent protein (GFP). The GFP antibody signal can be observed in the leaf blade of transgenic lines, but no signal was observed in wild type (WT) rice (Figure 2a,d), suggesting the high specificity of the GFP antibody. This signal was mainly in the phloem region of vascular bundles under Mg sufficient condition (Figure 2b,e). However, under Mg deficient condition, the signals were observed not only in the phloem region, but also xylem parenchyma cells and mesophyll cells in leaf blades (Figure 2c,f), indicating that Mg deficiency alters the tissue expression pattern of *OsMGT1* in leaves.

### 2.3. Knockout of OsMGT1 Resulted in Higher Sensitivity to Mg Deficiency

To investigate the physiological role of *OsMGT1* in rice under Mg deficient condition, the WT and two independent *OsMGT1* knockout lines were grown hydroponically with different concentrations of Mg supply. Under sufficient Mg (250 µM) supply, the growth was the same between the WT and two mutants (Figure 3a). However, under insufficient Mg (10 and 50 µM) supply, two mutants showed growth retardation compared with WT (Figure 3a), presenting a 20%–40% decrease in dry weight (Figure 3b). Furthermore, Mg deficient phenotypes such as leaf inclination and chlorosis were observed more evidently in two mutants (Figure 3c,d). The spectral plant analysis diagnostic (SPAD) values in fully expanded leaf of the mutants were remarkably lower than that of WT (Figure 3e). The angles of lamina joint in mutants became significantly larger than that in WT (Figure 3f). On the other hand, mineral analysis showed that Mg content was increased in both WT and mutants with increasing external Mg supply (Figure 3g). Nevertheless, two mutants showed much lower Mg content than WT at each Mg concentration (Figure 3g). All of these results indicate that *OsMGT1* plays an important role in rice growth under Mg-limited conditions.

### 2.4. Excessive Ca Aggravated Mg Deficiency in osmgt1 Mutants

To test the effect of Ca on *OsMGT1*-mediated Mg transport, the WT and two mutants were grown in the nutrient solution containing normal Ca (180 µM) or high Ca (1800 µM) concentrations in the presence of low Mg (10 µM). Our results showed that high Ca did not affect the growth of WT, but significantly inhibited the growth of *osmgt1* mutants under low-Mg conditions (Figure 4a). The parameters including plant height, SPAD value, dry weight and Mg content of the mutants were reduced more evidently by high Ca supply (Figure 4b–e), which suggests that the sensitivity to Mg deficiency in *osmgt1* mutants can be aggravated by excessive Ca. By contrast, high Ca supply has little influence on these parameters in WT (Figure 4b–e), suggesting that *OsMGT1* is also required for rice growth under Ca-aggravated Mg deficiency.

### 2.5. Overexpression of OsMGT1 Promoted Rice Growth Under Low Mg Stress

We generated three independent transgenic lines overexpressing *OsMGT1* (Figure 5a), in order to explore its genetic potential. We compared the WT and three overexpression lines hydroponically in the nutrient solution containing deficient (10 µM) and sufficient (250 µM) Mg concentrations. Our results showed that overexpression of *OsMGT1* resulted in better growth of rice plants under low-Mg conditions (Figure 5b), which was achieved by the increased Mg content and dry weight in transgenic lines (Figure 5c,e). By contrast, overexpression of *OsMGT1* did not improve the rice growth under Mg sufficient conditions (Figure 5d,f).

## 3. Discussion

*MRS2/MGT* family members have been identified as main Mg transporter genes in both prokaryote and eukaryote [21,26,27,28,29]. However, unlike other elemental transporter genes, the expression of these genes is rarely up-regulated by Mg deficiency [46,47,48,49]. So far, only *AtMGT6* in Arabidopsis roots has been revealed to be quickly induced by Mg deficiency, which is required for root Mg uptake [36]. In this study, we investigated the expression of all the *OsMGTs* in response to Mg deficiency in rice. Unexpectedly, none of them was able to be significantly induced by Mg deficiency for 7 days in rice roots (Figure 1a), suggesting that Mg transport systems in rice and Arabidopsis might be differently regulated. Notably, among nine members, only *OsMGT1* in shoots were highly induced by Mg deficiency (Figure 1a). Furthermore, the induction of *OsMGT1* by Mg deficiency is achieved by altering the tissue expression patterns in shoots. Under Mg sufficient condition, *OsMGT1* is only expressed in the phloem region of shoot vascular bundle (Figure 2b,e). However, the expression was remarkably enhanced in xylem parenchyma and leaf mesophyll cells by Mg deficiency (Figure 2c,f). Since OsMGT1 is a plasma membrane-localized Mg transporter, we speculate that rice is able to enhance Mg acquisition by facilitating both xylem Mg unloading and Mg import into leaf mesophyll cells by OsMGT1, in order to overcome Mg deficiency in shoots. However, *OsMGT1* in roots can be highly induced by Al toxicity and salt stress [25,50,51,52], suggesting that it has a more important role in the resistance to abiotic stresses than root Mg uptake. Indeed, knockout of *OsMGT1* only reduces one-third amount of Mg in rice roots [25,50,51,52], suggesting that there are other transporters mediating Mg uptake in rice. In bacteria, the repressible Mg uptake is mediated by the MgtA and MgtB protein [53]. Unlike CorA, MgtA and MgtB are P-type ATPases that mediate Mg influx [54,55]. However, whether P-type ATPases are involved in Mg uptake in plants needs to be further clarified.

Comparison of WT and *osmgt1* mutants revealed that the WT grew better than two *osmgt1* mutants under low-Mg conditions (Figure 3a). The increased sensitivity to Mg deficiency in mutants was due to the much lower Mg content in plants (Figure 3g), which resulted in much severer chlorosis in the leaves and larger inclination in the lamina joint (Figure 3d,f). On the other hand, overexpression of *OsMGT1* improved rice growth under low-Mg conditions, which is accompanied with increased dry weight and Mg content in overexpression lines (Figure 5c,e). These results indicate that *OsMGT1* is required for the tolerance to Mg deficiency, and that it can be utilized for molecular breeding of low-Mg tolerant plants in the future.

Considering that Mg induced expression of *OsMGT1* can be further enhanced by excessive Ca supply (Figure 1d), we compared WT and *osmgt1* mutants under excessive Ca conditions. Interestingly, excessive Ca aggravated Mg deficiency in *osmgt1* mutants, but not in WT (Figure 4a). Consistent with this phenotype, the plant height, SPAD value, dry weight and Mg content were decreased more evidently in *osmgt1* mutants (Figure 4b–e). It is well-known that Ca has an inhibiting effect on Mg uptake, through competition for ion channels that directly inhibit Mg transport, or apoplastic binding sites that indirectly inhibit Mg transport [25,50,51,52]. Since MRS2/MGT transporters own unique structure characteristics that have high selectivity to Mg^2+^ [31,32], it is unlikely that *OsMGT1* also has a high affinity to Ca^2+^. One possibility is that exogenously addition of Ca competitively reduced Mg apoplastic binding, leading to much lower Mg uptake and more severe Mg deficiency in *osmgt1* mutants (Figure 4). By contrast, the WT is able to alleviate these stresses from low Mg and high Ca by upregulating *OsMGT1*. Therefore, our results suggest that upregulation of *OsMGT1* by excessive Ca is an indirect response for rice to survive under a much more severe Mg deficient condition that is caused by excessive Ca.

Since OsMGT1 has diverse roles in abiotic stresses, a question remains as to whether OsMGT1 is able to transport other metals, such as Al, Na or Ca? However, Al^3+^ and Na^+^ have different ionic valency to Mg^2+^, and Ca^2+^ shows a different hydrated radius to that of Mg^2+^. It is unlikely that OsMGT1 shares equal affinity with these four metals. Although direct evidence is still needed to clarify the permeability of OsMGT1 to other metals, we speculate that Mg has strong interactions with these three metals, which are not through channel or transporter competition. Mg^2+^ competes with Al^3+^ for cellular oxygen donor compounds, competes with Ca^2+^ for apoplastic binding sites, and facilitates Na^+^ xylem retrieval by activating Na transporter OsHKT1;5.

## 4. Materials and Methods

### 4.1. Plant Materials and Growth Conditions

Two *Tos-17* insertion lines, NF0595 (*osmgt1-1*) and NE4528 (*osmgt1-2*) for *OsMGT1*, were obtained from the Rice Genome Resource Center in Japan. The homozygous lines were screened by PCR using specific primers as described in Chen, et al. [35]. In order to construct transgenic rice for overexpression of *OsMGT1*, the ORF of *OsMGT1* was amplified by PCR using primers Ubi: OsMGT1F and Ubi: OsMGT1R (Appendix A). The resulted fragment was cloned into *Kpn*I/*Bam*HI sites of pCAMBIA2300 vector driven by a *ubiquitin* promoter. The construct was introduced into the calluses of rice (cv Nipponbare) via *Agrobacterium tumefaciens*-mediated transformation [56].

Seeds of wild-type rice (cv Nipponbare), two knockout mutants, and overexpression lines were soaked in deionized water at 30 °C in the dark for 2 days and then transferred to a net floating on a 0.5 mM CaCl_2_ solution for 2 days, quarter strength Kimura B solution for 3 days, and half-strength Kimura B solution for 7 days as described in Yamaji and Ma [57]. The solution was renewed once every 2 days.

### 4.2. RNA Isolation and Gene Expression Analysis

In order to explore the effect of Mg deficiency on the expression of *OsMGTs*, a portion of 2-week-old rice seedlings were put into nutrient solution without Mg every day. After 7 days, the roots and shoots of all samples were separately harvested at the same time. After that, the additional 7 days Mg deficient seedlings were resupplied with 250 µM Mg for an additional day, and the samples including leaf blade, leaf sheath and node were separately harvested. In order to investigate the effect of excessive Ca on the expression of *OsMGT1*, 2-week-old rice seedlings were grown in the nutrient solution containing different concentrations of CaCl_2_ (180 µM and 1800 µM) in the presence of 10 µM Mg for 7 days, and the shoots were harvested.

Total RNA from rice tissues was extracted using the TransZol Up Plus RNA Kit (TransGen Biotech, Beijing, China). Half microgram of total RNA was used for first-strand cDNA synthesis using a TransScript One-Step gDNA Removal and cDNA Synthesis Super Mix (TransGen Biotech, Beijing, China) following the manufacturer’s instructions. The gene expression level was determined by real-time RT-PCR using the primers with TransStart Top Green qPCR Super Mix (TransGen Biotech, Beijing, China) on LightCycler 96 Real-Time PCR (Roche, Basel, Switzerland). Primers for real-time RT-PCR were listed in Appendix A. *OsActin* was used as an internal control. Normalized relative expression was calculated by the ΔΔCt method.

### 4.3. Immunohistological Analysis of OsMGT1

The transgenic plants carrying *promotorOsMGT1:GFP* were generated as described in Chen, et al. [38]. The seedlings were grown hydroponically in nutrient solution with or without Mg for 7 days. The middle parts of the youngest fully expanded leaves were sampled for immunostaining according to the method modified from Chen, et al. [58]. The samples were fixed in a solution containing 4% (*w*/*v*) paraformaldehyde, 60 mM sucrose and 50 mM sodium cacodylate for 2 h, and then embedded in 5% agar. The samples were sectioned to 60 µm thickness and incubated in 10 mM PBS containing 0.3% (*v*/*v*) Triton X-100 for 2 h. The slides were then incubated with GFP antibodies (anti-green fluorescent protein, Thermo Fisher Scientific, Somerset, NJ, USA) and subsequently with secondary antibodies (Alexa Fluor 555, Molecular Probes, Eugene, OR, USA). We observed the sections with a laser scanning confocal microscope (LSM880, Carl Zeiss, Oberkochen, Germany).

### 4.4. Phenotypic Analysis

To compare the sensitivity to Mg deficiency between WT and *osmgt1* mutants, 2-week-old seedlings of both WT (cv Nipponbare) and two *OsMGT1* knockout lines were grown in the nutrient solution containing 10 µM, 50 µM or 250 µM Mg. After 45 days, the rice seedlings were photographed. The SPAD values of the youngest fully expanded leaves were measured by a chlorophyll meter (SPAD-502 Plus, Konica Minolta, Tokyo, Japan) and the angles of lamina joint in each leaf were recorded by protractor. The plants were sampled after the roots were washed with 5 mM CaCl_2_ for three times to remove the apoplastic cations. The dry weight of the plants was weighed after being dried in a 75 °C oven for 2 days. Mg was determined by ICP-MS as described below.

The effect of excessive Ca on Mg deficiency was investigated by exposing 2-week-old WT and two *OsMGT1* knockout lines to a nutrient solution containing 180 µM or 1800 µM CaCl_2_ in the presence of 10 µM MgCl_2_ for 30 days. The rice seedlings were photographed. The plant height was measured by a ruler. The SPAD values of the youngest fully expanded leaves were measured using a chlorophyll meter. The dry weight of the plants was weighed after dried in 75 °C oven for 2 days. Mg was determined by ICP-MS as described below.

To investigate whether the tolerance to Mg deficiency is altered by overexpression of *OsMGT1* in rice, 2-week-old seedlings of WT and three overexpression lines were exposed to a nutrient solution containing 10 µM or 250 µM Mg. After 30 days, the rice seedlings were photographed. The dry weight was weighed after dried in a 75 °C oven for 2 days. Mg was determined by ICP-MS as described below.

### 4.5. Mg Determination in Plant Tissues

After harvest, the tissues were dried at 75 °C for 2 days to constant weight and then were subjected to digestion with concentrated HNO_3_ (68%) at a temperature of up to 140 °C. Mg concentration in the digested solution was determined by inductively coupled plasma-mass spectrometry (ICP-MS 7900, Agilent Technologies, Santa Clara, CA, USA). The Mg content was calculated based on Mg concentration and dry weight.

## 5. Conclusions

Taken together, our results conclude that OsMGT1 is required for the resistance to Mg deficiency in rice through facilitating both Mg transfer in xylem and Mg import in mesophyll cells.

## Figures and Tables

**Figure 1 ijms-20-00207-f001:**
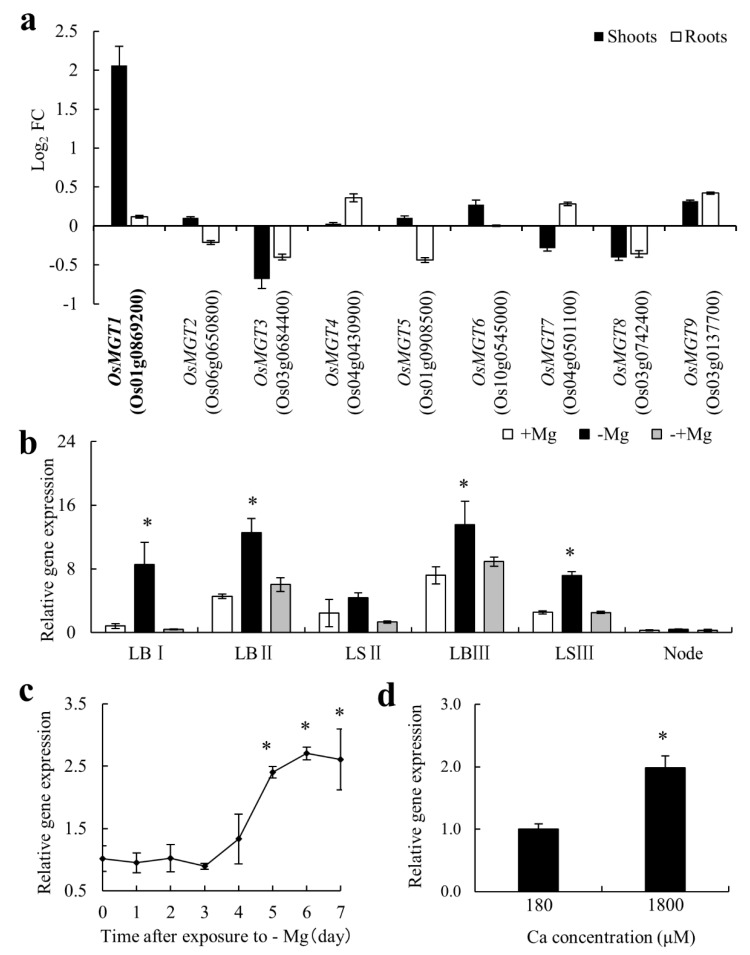
Gene expression pattern of *OsMGT1* in response to Mg deficiency. Gene expression of all *OsMGT_S_* family members in both shoots and roots (**a**). FC, fold change of induced expression. Effect of Mg sufficiency (+Mg), Mg deficiency (−Mg) and resupply (−+Mg) on the expression of *OsMGT1* in shoot tissues (**b**), including leaf blade (LB), leaf sheath (LS) and node. I to III is from young to old. Time-dependent expression of *OsMGT1* in shoots after exposure to −Mg (**c**). Effect of excessive Ca on the expression of *OsMGT1* under −Mg condition (**d**). The expression level was determined by real-time RT-PCR. *OsActin* was used as an internal standard. Data are means ± SD (*n* = 3). The asterisk indicates significantly different (*p* ≤ 0.05 by Tukey’s test).

**Figure 2 ijms-20-00207-f002:**
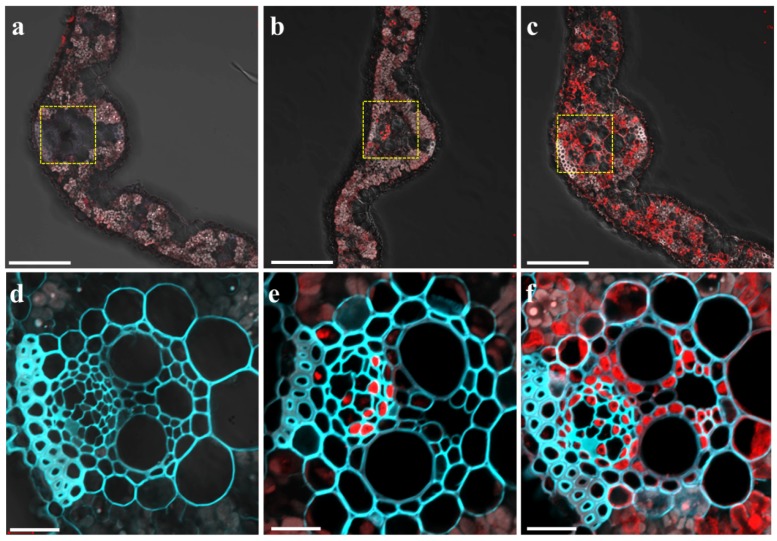
Tissue-specific and Mg-responsive expression of *OsMGT1*. Immunostaining with an anti-GFP was performed in the leaf blade of wild-type rice (**a**,**d**) and *pOsMGT1-GFP* transgenic rice under +Mg (**b**,**e**) and −Mg conditions (**c**,**f**). (**d**–**f**) are magnified images of yellow-dotted areas in (**a**–**c**) respectively. The red color represents the signal from the GFP antibody and cyan represents the signal from cell wall autofluorescence. Bars = 100 µm (**a**–**c**) and 20 µm (**d**–**f**).

**Figure 3 ijms-20-00207-f003:**
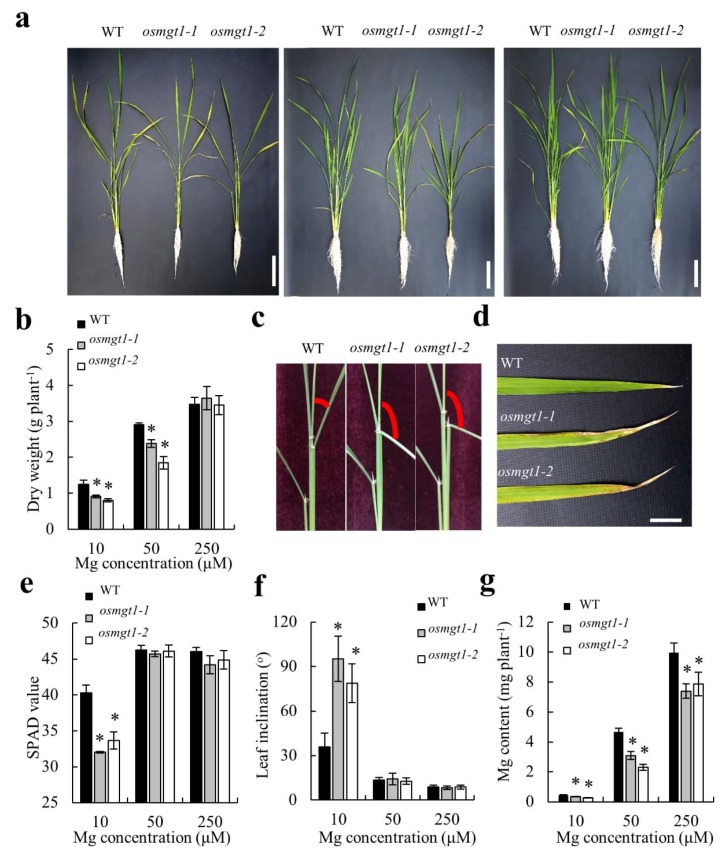
Sensitivity of *OsMGT1* knockout lines to Mg deficiency. Seedlings of both of WT and two *OsMGT1* knockout lines were grown with different Mg concentrations (**a**). Growth conditions (**a**). Left, 10 µM; middle, 50 µM; right, 250 µM. Dry weight (**b**), leaf chlorosis (**d**), spectral plant analysis diagnostic (SPAD) value (**e**), Mg content (**g**) and leaf inclination (**c**,**f**). Data are means ± SD (*n* = 3). The asterisk shows a significant difference compared with WT (*p* ≤ 0.05 by Tukey’s test). Bars = 10 cm (**a**) and 2 cm (**d**).

**Figure 4 ijms-20-00207-f004:**
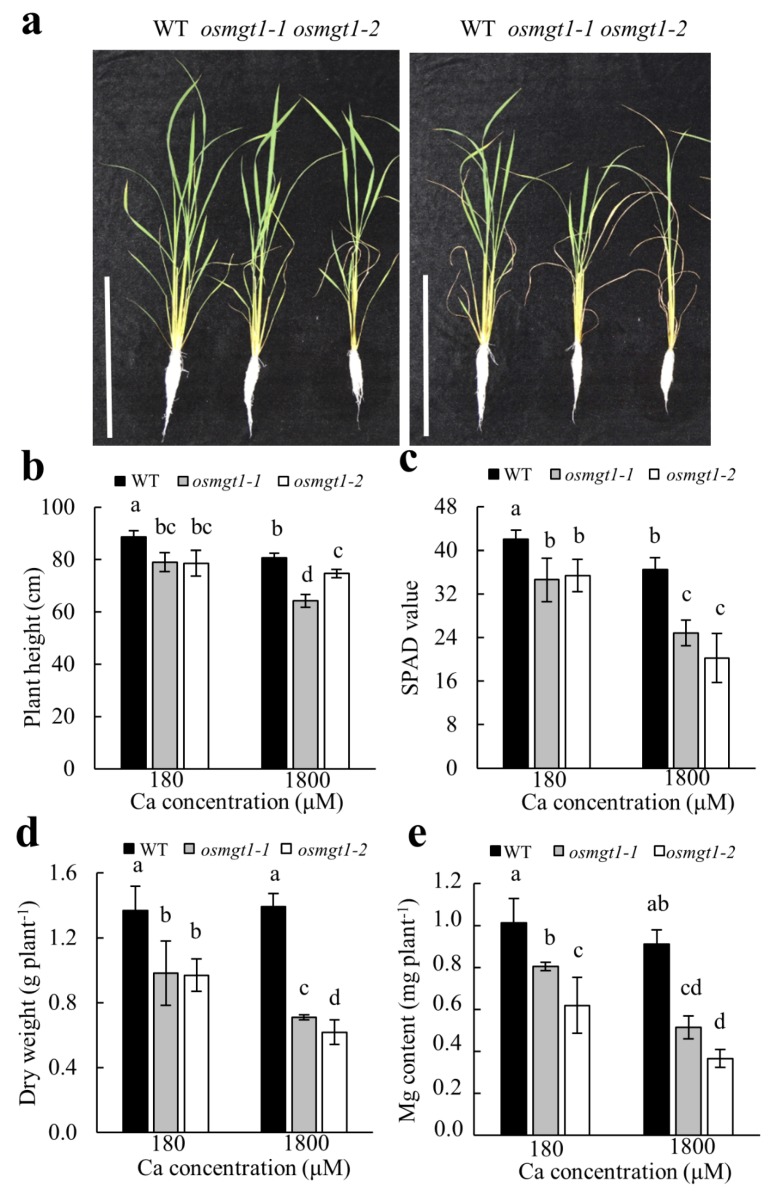
Aggravation of Mg deficiency by excessive Ca in *OsMGT1* knockout lines. Seedlings of both WT and two *OsMGT1* knockout lines were exposed to normal (180 µM) or high (1800 µM) Ca concentration in the presence of low Mg (10 µM). Growth conditions (**a**). Left, 180 µM; right, 1800 µM (**a**). Plant height (**b**), SPAD value (**c**), dry weight (**d**) and Mg content (**e**). Data are means ± SD (*n* = 3). Means with different letters are significantly different (*p* ≤ 0.05 by Tukey’s test). Bar = 30 cm.

**Figure 5 ijms-20-00207-f005:**
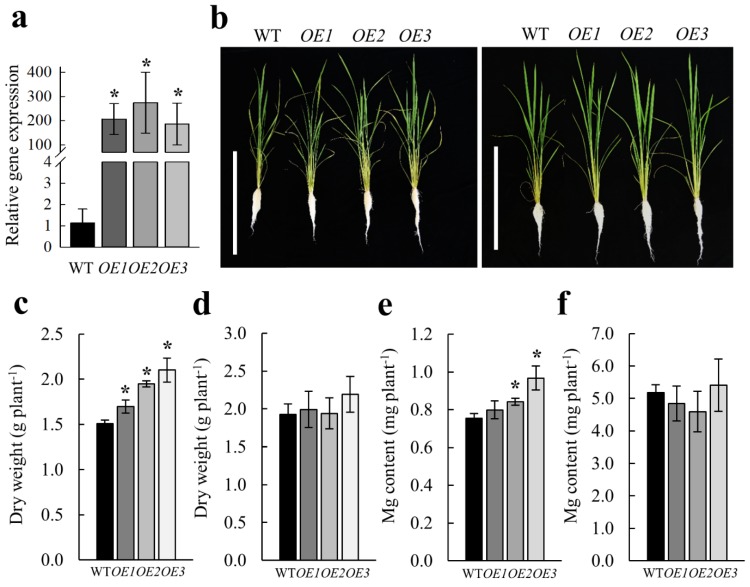
Overexpression of *OsMGT1* improved rice growth under low-Mg conditions. The WT and three overexpressing lines (*OE1*, *OE2* and *OE3*) were grown with deficient (10 µM) or sufficient (250 µM) Mg supply. Relative expression level of *OsMGT1* in three overexpressing lines (**a**). The expression level was determined by real-time RT-PCR. *OsActin* was used as an internal standard. Growth conditions (**b**). Left, 10 µM; right, 250 µM. Dry weight (**c**,**d**) and Mg content (**e**,**f**) under deficient and sufficient Mg supply. Data are means ± SD (*n* = 3). The asterisk shows a significant difference compared with WT (*p* ≤ 0.05 by Tukey’s test). Bar = 50 cm.

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
