# Peer review of "OsMGT1 Confers Resistance to Magnesium Deficiency By Enhancing the Import of Mg in Rice"

_ijms, 2019, doi:10.3390/ijms20010207_

Round 1

Reviewer 1 Report

This is a very well written and interesting report looking at the additional function of OsMGT1 in shoot under the magnesium (Mg) deficiency condition. The authors found that the high expression of OsMGT1 in shoot under the low Mg condition contribute to the resistance to Mg deficiency thorough increasing Mg transport.

I recommend adding some discussion that will help the reader for better understanding of this manuscript. OsMGT1 was Mg uptake transporter in rice root (Chen et al. 2012). To obtain enough Mg under the Mg insufficient condition, the expression of OsMGT1 should be upregulated in both the root and the shoot. However, OsMGT1 was upregulated only in the shoot but not in the root. Authors have to discuss more for about this very interesting point.

Minor comments:

Page 2, line 93

fifth instead of fourth

Page 3, Figure 1. a

Gene name should be italics

Page 3, Figure 1. b

Generally, leaf number from young to old should be presented by roman numerals, like LBI, LBII.

Page 3, Figure 1. c

The expression of OsMGT1 in +Mg condition should be presented in Figure 1c.

Page 4, line 110

(Fig. 2a, b) should be (Fig. 2a, d)

Page 6, line 149

for instead of from

Page7, line 162-

The expression levels of OsMGT1 ox lines should be presented or mentioned.

Author Response

This is a very well written and interesting report looking at the additional function of OsMGT1 in shoot under the magnesium (Mg) deficiency condition. The authors found that the high expression of OsMGT1 in shoot under the low Mg condition contribute to the resistance to Mg deficiency thorough increasing Mg transport.

I recommend adding some discussion that will help the reader for better understanding of this manuscript. OsMGT1 was Mg uptake transporter in rice root (Chen et al. 2012). To obtain enough Mg under the Mg insufficient condition, the expression of OsMGT1 should be upregulated in both the root and the shoot. However, OsMGT1 was upregulated only in the shoot but not in the root. Authors have to discuss more for about this very interesting point.

Thank you very much for your constructive comments. It is interesting that OsMGT1 can be remarkably up-regulated by -Mg in shoots but not in roots. Since OsMGT1 is a plasma membrane-localized Mg transporter, we speculate that rice is able to enhance Mg acquisition through facilitating both xylem Mg unloading and Mg import into leaf mesophyll cells by OsMGT1, in order to overcome Mg deficiency in shoots. However, OsMGT1 in roots can be induced by Al toxicity and salt stress, suggesting its more important role in the resistance to abiotic stresses than Mg uptake in roots. Indeed, knockout of OsMGT1 only reduces one third amount of Mg in rice roots (Chen et al., 2012; 2017), suggesting that there are other Mg-responsive transporters mediating Mg uptake in rice. We have added some discussions into the revised manuscript.

Minor comments:

Page 3, Figure 1. c

The expression of OsMGT1 in +Mg condition should be presented in Figure 1c.

We are sorry for our unclear description. For 7-d time-course experiment, we put a portion of rice seedlings to -Mg nutrient solution every day. After 7 days, we sampled them at the same time to avoid the period expression difference. Thus, the seedlings with -Mg for 0 days is actually a +Mg control. We have revised the method in the manuscript.

Page7, line 162-

The expression levels of OsMGT1 ox lines should be presented or mentioned.

We have added the expression data of OsMGT1 ox lines into Fig. 5

Page 2, line 93

fifth instead of fourth

Page 3, Figure 1. a

Gene name should be italics

Page 3, Figure 1. b

Generally, leaf number from young to old should be presented by roman numerals, like LBI, LBII.

Page 4, line 110

(Fig. 2a, b) should be (Fig. 2a, d)

Page 6, line 149

for instead of from

Thanks for all these important comments. We have revised the manuscript according to your suggestions. 

Reviewer 2 Report

This paper is analyzing a mutant rice line under different conditions.  The work shown is well presented and the organization is sound.   Papers in 2012 and 2017 have documented different phenotypes of this SAME mutant line.  The field would be improved if the authors characterize the transport properties - as the paper stands now much of the work is speculation based on localization and induction of expression (or lack of).  The relationship between Mg/Al/Ca/Na requires clarification beyond phenotype analysis.

Author Response

This paper is analyzing a mutant rice line under different conditions.  The work shown is well presented and the organization is sound.   Papers in 2012 and 2017 have documented different phenotypes of this SAME mutant line.  The field would be improved if the authors characterize the transport properties - as the paper stands now much of the work is speculation based on localization and induction of expression (or lack of).  The relationship between Mg/Al/Ca/Na requires clarification beyond phenotype analysis.

Thank you very much for your suggestions. We deeply agree with your ideas. We had previously thought that OsMGT1 also has the transport affinities to Al, Ca or Na, so we test its transport activity in yeast (CM66 and mrs2) and oocyte systems. Unfortunately, OsMGT1 protein cannot be expressed well in these systems for some unknown reason. Then we used stable isotope 25Mg to investigate Mg uptake ability in WT and osmgt1 mutants. WT exhibited much higher 25Mg uptake ability than mutants (Chen et al., 2012), and Mg concentration in WT roots also was higher than that in mutants (Chen et al., 2017). Except for Mg, we found that there was no significant difference in other elements, including Al, Ca and Na in the roots, suggesting the specificity of OsMGT1 to Mg. Since Al3+ and Na+ have different ionic valency with Mg2+, and Ca2+ shared different hydrated radius with Mg2+, it is unlikely that OsMGT1 share equal affinity to these four metals. Although direct evidences are still needed to clarify the permeability of OsMGT1 to other metals, we speculate that Mg has strong interactions with these three metals, which is not through channel or transporter competition. Mg2+ competes with Al3+ for cellular oxygen donor compounds, competes with Ca2+ for apoplastic binding sites and facilitates Na xylem retrieval by activating Na transporter OsHKT1;5.

We have added some discussions into the revised manuscript.

Reviewer 3 Report

This manuscripts provides some insight into the role of OsMGT1 in rice. The authors nicely presented the genetic and physiological studies. However, I would appreciate if they provide the results on some other nutrients - their uptake and interference. There are literture that Mg can interfere the uptake of other nutrients.

English language could be improved.

Please improve the discussion and cite recent and relavent references.

Author Response

This manuscripts provides some insight into the role of OsMGT1 in rice. The authors nicely presented the genetic and physiological studies. However, I would appreciate if they provide the results on some other nutrients - their uptake and interference. There are literture that Mg can interfere the uptake of other nutrients. 

Thank you very much for your comments and suggestions. We have listed the uptake data of other nutrients (K, P, Ca, Na, Cu, Mn, Fe, Zn) in WT and mutants (see data below). However, we could not find any significant difference in other nutrients between WT and mutants. We guess that Mg interferes the uptake of other nutrients through other mechanisms but not by OsMGT1 transporter.

English language could be improved.

We have asked a native speaker to correct the gamma and spelling mistakes in the text.

Please improve the discussion and cite recent and relavent references.

We have made some improvements on discussions and cited some recent references.

Reviewer 4 Report

Presented article is an further investigation on OsMGT1, gene that was already subject of intensive study from same of the authors of this work. In this study, OsMGT1 was identified as an Mg transporter conferring resistance to Mg deficiency.

The work is well written and the figures present the results on the proper way.

To my point of view, only Fig 5A must be revised. As the expression in wild type plants is not presented it is difficult to understand if the transformants are really over expressing. It is not clear what the asterisk above the bars on Fig5A represent as there are no explanation in the legend.

Author Response

Presented article is an further investigation on OsMGT1, gene that was already subject of intensive study from same of the authors of this work. In this study, OsMGT1 was identified as an Mg transporter conferring resistance to Mg deficiency.

The work is well written and the figures present the results on the proper way.

To my point of view, only Fig 5A must be revised. As the expression in wild type plants is not presented it is difficult to understand if the transformants are really over expressing. It is not clear what the asterisk above the bars on Fig5A represent as there are no explanation in the legend.

Thank you very much for your comments and suggestions. Fig. 5A has been revised, and showed the expression level of WT and overexpression lines. The asterisk shows a significant difference compared with WT (P < 0.05 by Tukey’s test).

Round 2

Reviewer 2 Report

As a "major" revision- the authors have added some text to explain the lack of transport data.

The work does not show Mg transport (as the title suggests) but infers transport.

Author Response

Thanks for your comments. We have revised the title according to your suggestions, and substitute the word “import” for “transport”. Accordingly, we have checked all words throughout the paper to infer transport but not to show Mg transport.